# Early Prediction and Monitoring of Treatment Response in Gastrointestinal Stromal Tumors by Means of Imaging: A Systematic Review

**DOI:** 10.3390/diagnostics12112722

**Published:** 2022-11-07

**Authors:** Ylva. A. Weeda, Gijsbert M. Kalisvaart, Floris H. P. van Velden, Hans Gelderblom, Aart. J. van der Molen, Judith V. M. G. Bovee, Jos A. van der Hage, Willem Grootjans, Lioe-Fee de Geus-Oei

**Affiliations:** 1Department of Radiology, Leiden University Medical Center, 2333 ZA Leiden, The Netherlands; 2Department of Medical Oncology, Leiden University Medical Center, 2333 ZA Leiden, The Netherlands; 3Department of Radiology, Section of Abdominal Radiology, Leiden University Medical Center, 2333 ZA Leiden, The Netherlands; 4Department of Pathology, Leiden University Medical Center, 2333 ZA Leiden, The Netherlands; 5Department of Surgical Oncology, Leiden University Medical Center, 2333 ZA Leiden, The Netherlands; 6Biomedical Photonic Imaging Group, University of Twente, 7522 NB Enschede, The Netherlands; 7Department of Radiation Science & Technology, Technical University of Delft, 2629 JB Delft, The Netherlands

**Keywords:** gastrointestinal stromal tumor, prediction, response monitoring, FDG-PET, radiomics, tomography, X-ray computed, personalized medicine

## Abstract

Gastrointestinal stromal tumors (GISTs) are rare mesenchymal neoplasms. Tyrosine kinase inhibitor (TKI) therapy is currently part of routine clinical practice for unresectable and metastatic disease. It is important to assess the efficacy of TKI treatment at an early stage to optimize therapy strategies and eliminate futile ineffective treatment, side effects and unnecessary costs. This systematic review provides an overview of the imaging features obtained from contrast-enhanced (CE)-CT and 2-deoxy-2-[^18^F]fluoro-D-glucose ([^18^F]FDG) PET/CT to predict and monitor TKI treatment response in GIST patients. PubMed, Web of Science, the Cochrane Library and Embase were systematically screened. Articles were considered eligible if quantitative outcome measures (area under the curve (AUC), correlations, sensitivity, specificity, accuracy) were used to evaluate the efficacy of imaging features for predicting and monitoring treatment response to various TKI treatments. The methodological quality of all articles was assessed using the Quality Assessment of Diagnostic Accuracy Studies, v2 (QUADAS-2) tool and modified versions of the Radiomics Quality Score (RQS). A total of 90 articles were included, of which 66 articles used baseline [^18^F]FDG-PET and CE-CT imaging features for response prediction. Generally, the presence of heterogeneous enhancement on baseline CE-CT imaging was considered predictive for high-risk GISTs, related to underlying neovascularization and necrosis of the tumor. The remaining articles discussed therapy monitoring. Clinically established imaging features, including changes in tumor size and density, were considered unfavorable monitoring criteria, leading to under- and overestimation of response. Furthermore, changes in glucose metabolism, as reflected by [^18^F]FDG-PET imaging features, preceded changes in tumor size and were more strongly correlated with tumor response. Although CE-CT and [^18^F]FDG-PET can aid in the prediction and monitoring in GIST patients, further research on cost-effectiveness is recommended.

## 1. Introduction

Gastrointestinal stromal tumors (GISTs) are rare mesenchymal neoplasms affecting the entire gastrointestinal tract and are presumed to originate from the interstitial cells of Cajal [1,2]. About 80–90% of GISTs harbor kinase-activating mutations in either receptor tyrosine kinase protein (KIT) or platelet-derived growth factor receptor α (PDGRF-α) [3,4]. Complete surgical excision remains the only curative treatment option for GIST patients. Since GISTs are generally insensitive to radio- and chemotherapy, non-surgical treatment is limited to tyrosine kinase inhibitor (TKI) therapy. This targeted molecular therapy is part of routine clinical practice for unresectable and metastatic disease [5,6].

Adjuvant TKI treatment is used in high-risk GISTs to improve survival [7]. Unfortunately, due to the varying aggressive nature of GISTs, about one-third of the patients will relapse within three years after surgery with curative-intent [8]. For localized disease, TKI treatment can be given to attain size reduction of the primary tumor and improve chances of complete resection while maintaining an acceptable risk of complications [9,10]. About 20–25% of patients do not benefit from the neoadjuvant TKI treatment, as no complete or partial response is observed [11,12]. The rarity and complex biological nature of this disease, makes it difficult to differentiate between good and poor responders. For example, GISTs harboring a KIT exon 11 mutation have a good response to TKI treatment, whereas the same treatment is less effective in tumors with KIT exon 9 mutations [13]. Additionally, progressive disease is common during long-term TKI treatment due to acquired treatment resistance [14,15].

In the era of personalized medicine, it is of utmost importance to evaluate the efficacy of TKI treatment at an early stage in order to optimize therapy strategies and protect patients from futile ineffective treatment, unnecessary side-effects and healthcare costs. Contrast-enhanced computed tomography (CE-CT) and 2-deoxy-2-[^18^F]fluoro-D-glucose ([^18^F]FDG) PET/CT are considered useful for diagnosis and response monitoring in GIST patients. The imaging modalities offer information on tumor morphology, perfusion characteristics, as well as tumor glucose metabolism [7]. However, optimal use of imaging for predicting and monitoring TKI treatment in patients with GIST is still a subject of debate. This systematic review aims to elucidate the added value of CE-CT and [^18^F]FDG PET/CT imaging in the *prediction of response* and *early response monitoring* of TKI treatment in localized and advanced GISTs.

## 2. Methods

### 2.1. Search Strategy

From 29 April 2022 to 24 June 2022, the databases of PubMed, Web of Science, the Cochrane Library and Embase were systematically screened using predefined search queries (Appendix A). The following terms and their corresponding synonyms were included: “gastrointestinal stromal tumors”, “(neo)adjuvant”, “TKI treatment” and “FDG-PET” and “Tomography, X-ray Computed” imaging. The search queries are wide-ranging and seek to cover the aspect of both response monitoring and prediction by including ‘monitoring’ and ‘prediction models’ as well as ‘radiomics’ and ‘prognostics’. In addition to these search terms, other terms, such as ‘patient selection’ and ‘personalized medicine’, were also added, since these articles presumably covered the subject of TKI treatment evaluation and its efficacy in specific patient groups as well. The search strategy was implemented in consultation with an experienced research directorate, and access to the databases was granted by the Leiden University Medical Center.

### 2.2. Article Selection

Articles were screened and considered eligible for full-text assessment if the title or abstract mentioned (i) quantitative outcome measures to evaluate the efficacy of imaging features (ii) retrieved from CE-CT and/or [^18^F]FDG PET/CT imaging (iii) in predicting or monitoring (neo)adjuvant TKI treatment response (iv) in localized and advanced GISTs. Response monitoring is defined as the evaluation of disease over the course of treatment using multiple medical imaging time points. Predicting response, however, solely involves the use of baseline scans made prior to TKI treatment administration. Articles assessing the prognostic value of different clinical and imaging parameters (e.g., risk of recurrence and metastatic potential) that can guide TKI treatment duration or timing for specific patient groups were also included, since these findings may improve patient selection in the future. Exclusion criteria comprised non-English and non-human studies, reviews, guidelines, recommendations, editorials, conference papers and abstracts. Case reports and studies analyzing less than ten patients were also excluded. If the title and abstract did not contain sufficient information, full-text evaluation was used for judgement of relevance.

Subsequently, the articles were screened on full-text and excluded if they did not meet the previously mentioned inclusion criteria or if full-texts were not available. During this assessment, the focus was primarily on quantitative outcome measure(s) of studies. Outcome measures that were included in this analysis were correlations, associations, area under the curve (AUC), sensitivity, specificity and accuracy. 

Finally, the reference lists from included articles were screened to find additional articles on this topic. The articles were independently assessed by the first two authors (Y.A.W., G.M.K.) and in cases of discrepancy, consensus reading was performed to make a final decision that led to either inclusion or exclusion.

### 2.3. Quality Assessment

Articles using a radiomics pipeline were assessed through the radiomics quality score (RQS). The RQS is a scoring system that assigns points to a radiomics study based on specific criteria, where a maximum score of 36 points can be awarded. In this paper, the RQS is modified to focus on the methodological aspects of the included studies. The following criteria were omitted from the RQS, yielding a modified RQS (RQS_m_); ‘imaging at multiple time points’, ‘trial database registry’ and ‘multivariable analysis on non-radiomics features’, since they were considered less relevant for the quality of the obtained models [16]. The criteria from the RQS_m_ were also used to create a quality assessment tool to assess studies on non-radiomics prediction models and correlational research. Modifying the RQS_m_ for non-radiomics studies yielded the RQS_m,nonrad_. This RQS_m,nonrad_ had a maximum score of ten points (Appendix A). Articles were considered high quality if they reached a score above 50%. To assess applicability concerns and the risk of bias in articles covering the topic of monitoring, the Quality Assessment of Diagnostic Accuracy Studies Tool, Version 2 (QUADAS-2) was applied [17]. Articles on response monitoring were considered to have a high risk of bias or applicability concerns if two or more of the domains were scored as ‘high’ or ‘unclear’. Subsequently these articles were scored as low-quality. 

Quality assessment was performed by the first author (Y.A.W.). The quality score was not considered as an exclusion criterion, as the authors considered it important to review all relevant evidence [17,18,19]. 

### 2.4. Data Analysis

The eligible studies were categorized based on their topic concerning either response prediction or therapy monitoring. From each study, detailed information on the publication year, first author, patient groups, type of TKI treatment and imaging technique(s), was obtained. The specific CE-CT and [^18^F]FDG-PET imaging features and their corresponding conclusions on efficacy, along with the attributed quality score, were briefly summarized. In the results section, only studies that were considered to be high-quality, were analyzed in depth by clarifying conclusions on clinical relevance, discrepancies in results and insights on biological correlates.

#### 2.4.1. Response Prediction

In response prediction, imaging features from baseline/diagnostic CE-CT and [^18^F]FDG-PET/CT are retrieved to predict responder status, prior to TKI administration. Articles on this topic were divided into five categories: mutational status, proliferative activity, risk stratification, radiological response and prognosis. These categories were considered important, as they all influence treatment strategies. Clinical genotyping is essential for clinical decision making regarding neoadjuvant therapy, since the sensitivity and resistance towards TKI treatment in GISTs is dependent on the mutational status. In addition, patients with a high-risk GIST (and thus high proliferative activity) receive adjuvant TKI treatment for a period of three years to eliminate remaining disease and reduce chances of relapse [7]. Predicting whether patients will have a radiological response or a good prognosis at baseline could also aid the development of a more personalized TKI treatment.

#### 2.4.2. Therapy Monitoring

In therapy monitoring, one uses the visual and quantitative differences between baseline and follow-up scans to determine treatment response. The efficacy of CE-CT and [^18^F]FDG-PET are first discussed separately, followed by a qualitative comparison between both imaging modalities. 

## 3. Results

### 3.1. Search Strategy and Article Selection

The search query identified a total of 599 articles from the databases of PubMed, Web of Science, the Cochrane library and Embase. The study selection process led to a total of 90 articles eligible for analysis (Figure 1). Articles that were excluded based on imaging criteria included, for example, the use of radiotracers other than [^18^F]FDG [20]. Additionally, some articles discussed the use of molecular genotyping and DNA sequencing to predict or determine response and therefore did not involve the use of any imaging modality [21]. Other excluded articles discussed the efficacy of a specific TKI treatment but did not quantify the efficacy of imaging features in predicting or monitoring response [22,23]. Of the 90 eligible articles, 67 were concerning response prediction [24,25,26,27,28,29,30,31,32,33,34,35,36,37,38,39,40,41,42,43,44,45,46,47,48,49,50,51,52,53,54,55,56,57,58,59,60,61,62,63,64,65,66,67,68,69,70,71,72,73,74,75,76,77,78,79,80,81,82,83,84,85,86,87,88,89,90] and 23 discussed response monitoring [91,92,93,94,95,96,97,98,99,100,101,102,103,104,105,106,107,108,109,110,111,112,113].

### 3.2. Quality Assessment

Twenty-two articles discussed the use of radiomic models, and six out of 22 studies were of low quality (score < 50%). The mean RQS_m_ of the included articles was 13.5 (SD ± 2.60) out of 26 points. Low scores were mainly caused by a lack of transparency, biological correlates and gold standard comparison. Two articles received a score of 18 points (69.2%), which was the highest attributed score [70,88]. The forty-five studies on non-radiomic prediction models and correlational research scored an average RQS_m,nonrad_ of 3.91 (SD ± 1.23) out of ten points, where eighteen articles scored above 50.0%. This was mainly caused by the fact that only a few articles used gold standard comparison [31,35,46] or an undescribed test set to validate their results [44,48,68]. The results of the QUADAS-2 tool are graphically displayed in Figure 2. Eight articles on response monitoring had high risk of bias or concerns for applicability and were therefore scored as low-quality. High risk of bias was often introduced by using reference standards involving follow-up (e.g., progression free survival, overall survival, time-to-treatment failure). Concerns for applicability were mainly caused by a lack of reporting on patient characteristics. In this way, judgement on whether the included patients matched the review question was unclear.

### 3.3. Response Prediction

All articles on response prediction have been summarized in the Appendix A. Studies that were considered high-quality will be discussed in more detail.

#### 3.3.1. Mutational Status

The radiomic model of Starmans et al. was validated on unseen data and achieved an AUC, sensitivity and specificity of 0.51, 96.0% and 3.00% for predicting KIT mutation presence [81]. The model, based on portal venous radiomic features, requires further improvement in order to be clinically applicable. 

Three studies developed a model or nomogram based on radiomic features obtained from CE-CT imaging (arterial, venous and delayed phase) to predict the presence of KIT exon 11 mutations, which resulted in varying AUC outcomes, namely 0.57, 0.72 and 0.81 [75,76,81]. Deletions in exon 11 may indicate more aggressive tumor behavior, and for this reason, Liu et al. also assessed the efficacy of their model in predicting exon 11 deletion affecting codons 557–558 and achieved an AUC of 0.85 [76]. 

In clinical practice, patients with KIT exon 9 mutations often receive a high-dose imatinib regimen (800 mg) to improve progression-free survival (PFS). Yin et al. showed significantly greater tumor sizes and higher enhancement ratios (Hounsfield units (HU) for tumor parenchyma divided by the HUs of the erector spinae muscle) on portal venous CE-CT imaging compared to KIT exon 11 mutations. Using a 1.60 cut-off point, KIT exon 9 mutated small intestine tumors could be differentiated with an AUC, sensitivity and specificity of 0.76 and 86.7% and 98.5%, respectively. This threshold has, however, not been validated on independent validation data [67]. 

#### 3.3.2. Proliferative Activity

Since high-risk GISTs have a high proliferation rate, several studies attempted to link the mitotic index and Ki-67 proliferation index to imaging features in order to make a non-invasive assessment of expected tumor behavior. On CE-CT imaging, intralesional hypodensity and concurrent heterogeneous enhancement patterns were significantly more common in high-mitotic tumors (Figure 3 and Figure 4) [29,46]. Hypodensity was, in this case, defined as an area of low attenuation on portal venous phase CE-CT with HUs between 0 and 30 and when no HU increase (max 5 HUs) was observed between unenhanced and post-contrast images [46]. The changes in enhancement patterns were attributed to the principle of neovascularization. Tumors with high proliferative activity can induce the formation of hyperpermeable disorganized blood vessels and consequent development of necrosis [29,61]. Therefore, the supply and washout of contrast agent is affected, which has a direct impact on tumor enhancement patterns. 

A radiomic model using 42 quantitative and semantic imaging features (tumor location, first-order and texture radiomic features) retrieved from portal venous CE-CT imaging, differentiated high- from low-mitotic tumors with an AUC, sensitivity and specificity of 0.54, 27.0% and 75.0%, respectively [81]. Although on theoretical grounds CE-CT should be able to visualize poor neo-vasculature due to rapid tumor growth, no radiomic study has been able to establish this correlation. However, radiomic models predicting high Ki-67 proliferation index in localized and advanced GISTs achieved AUC values above 0.75 [77,88,89].

Comparison of studies investigating the relation between imaging and Ki-67 indices is complicated by the fact that different thresholds (e.g., 4%, 5%, 8% and 10%) for Ki-67 expression were used. Due to the small study sizes and heterogeneous outcomes with respect to Ki-67 indices, the true relationship between CE-CT imaging and proliferation has yet to be established.

#### 3.3.3. Risk Stratification

Research on the use of [^18^F]FDG-PET imaging features for risk stratification in GISTs is limited. In two studies, high metabolic tumor volume (MTV) and total lesion glycolysis (TLG) were predictive for high risk GISTs [25,35]. The use of quantitative imaging features showed improved predictive accuracy during follow-up when compared to a clinical reference standard (NIH modified criteria) [35]. Although these results suggest the added role of [^18^F]FDG-PET for risk stratification, there are only a few studies that investigated [^18^F]FDG-PET for this purpose. 

Larger tumor sizes, mixed or extra-luminal growth patterns, ill-defined tumor shape, presence of vessels feeding or vessels draining the tumor mass, necrosis and ulceration on CE-CT imaging were all associated with high-risk GISTs [44,53,58,60,63,64,68]. Of note, Wei et al. used the angle between the longest and shortest tumor diameter to quantify tumor shape. This parameter was able distinguish intermediate- and high-risk from low-risk GISTs more accurately when compared to using solely the longest diameter [58]. Heterogeneous enhancement patterns on portal venous phase CE-CT proved to be predictive for high-risk GISTs as well (Figure 5) [53]. Incomplete enhancement of the overlying gastric mucosa on arterial phase, was also significantly more common in high-risk gastric GISTs [51]. In a study by Tang et al., HUs of the arterial phase CE-CT were subtracted from the attenuation coefficients in the portal venous phase to derive quantitative features describing contrast enhancement. Using the subtraction CT, small regions of interest (ROIs) of 30–50 mm^2^, were placed in the most enhancing solid components of the tumors. The difference in HUs was significantly lower in high-risk gastric GISTs [53]. Additionally, the peak value of enhancement on CE-CT (arterial and portal venous phase) imaging was strongly correlated with risk [45]. Both articles suggest a rapid inflow of iodinated contrast agent in high-risk GISTs and thus the presence of permeable and leaky tumor vessels. The mean of the positive pixels (HU > 0) of the entire tumor volume on portal venous CT imaging was lower in high-risk GISTs [31]. This observation can be attributed to the presence of tumor necrosis, which was more commonly found in the high-risk group.

By contrast, Li et al. included gastric, intestinal and extra gastrointestinal tumors and did not find a significant difference in enhancement patterns between risk groups [43]. Although tumor enhancement has been established as a relevant factor in the risk stratification of GISTs, there are discrepancies in the results.

Machine learning used for the prediction of risk is extensively investigated with a total of twelve articles covering this topic [71,72,79,83,86,87]. All models achieved an AUC above 0.83 for predicting high-risk GISTs, with an average AUC of 0.87. In many of the models, texture radiomic features (gray level co-occurrence matrix (GLCM), neighboring gray-tone difference matrix (NGTDM) and gray run-length matrix (GRLM) and gray level size zoned matrix (GLSZM)) were used to develop the model. These texture features reflect enhancement patterns and inter-pixel relationships in a three-dimensional tumor volume. 

#### 3.3.4. Prediction of Radiological Response

There was one article attempting to predict radiological response using baseline imaging. Disease progression was in this case defined by the modified Choi criteria, which is currently one of the reference standards used for GIST response evaluation [114]. In this case, four textural portal venous features (features retrieved from GLCM, GLRLM and NGTDM) predicted disease progression with an AUC of 0.83 [32]. 

#### 3.3.5. Prognosis

Of the selected articles, two articles discussed the use of imaging features obtained from [^18^F]FDG-PET/CT imaging to predict PFS through detection of disease recurrence (locally and or development of distant metastases). They found significantly higher MTV and TLG values in patients with a lower PFS. In addition to quantitative [^18^F]FDG-PET imaging features, larger tumor sizes were also a significant factor contributing to lower PFS [25,35]. 

On CE-CT imaging, one study with a relatively large patient group (n = 143) observed that tumor sizes greater than 10 cm, ill-defined tumor outline and enhancing solid components contributed to a poor patient prognosis, as reflected by their overall survival (OS) [48]. The study by Jung et al. combined relevant predictive parameters (tumor location, ill-defined tumor outline and presence of feeding vessels) to create a nomogram. The nomogram was internally validated and achieved an AUC of 0.86 [37]. In addition to semantic CT features, Ekert et al. assessed the efficacy of four quantitative textural features from portal venous phase CT imaging to predict prognosis of GIST patients. This study found that high values for these texture features were all associated with poor PFS [32].

In another study, three-year recurrence-free survival (RFS) was predicted by a deep learning ResNet model based on features retrieved from arterial phase CT images. Results show that, using an internal validation cohort, a predictive model with an AUC of 0.91 was obtained [70]. Furthermore, Zheng et al. investigated whether the occurrence of liver metastasis in high risk GISTs could be predicted. They found that a model based on portal venous CT radiomic features reached an AUC and accuracy of 0.87 and 84.9% [90]. 

### 3.4. Therapy Monitoring

All articles on therapy monitoring have been summarized in the Appendix A. Studies that were considered high-quality will be discussed in more detail.

#### 3.4.1. CE-CT Imaging

Many articles discussed the use of the well-established Response Evaluation Criteria in Solid Tumors (RECIST 1.1) to assess tumor response. RECIST 1.1 is a method in which the sum of the longest diameter of (a maximum of 5) target lesions is used to evaluate treatment response. The RECIST scoring system categorizes patients into four types of response, namely complete response (disappearance of all lesions), partial response (≥30% reduction of the sum of the target lesions (SLD)), progressive disease (≥20% increase of the SLD compared to the smallest SLD ever measured) and stable disease (neither progressive disease nor partial response) [115]. Nonetheless, substantial tumor shrinkage is often not observed during effective TKI treatment. Subtle and moderate changes in tumor size may be more accurately quantified by means of volumetric measurements. This is shown by Schiavon et al., who showed that size changes in GIST liver metastases larger than 20% were more frequently detected by volumetric measurements compared to the RECIST 1.1 criteria [110]. By using solely one-dimensional measurements, one presumes tumors remain spherical and that response occurs equally along three orthogonal axes during TKI treatment. However, liver metastasis in GIST patients showed significant changes in morphology over the course of imatinib treatment, which was better reflected by an ellipsoid volumetric approach [109]. 

In addition to RECIST 1.1, Choi et al. proposed a new method (Choi criteria) by including treatment-related changes in portal venous CT tumor densities [95]. Suppression of vascular endothelial growth factor expression can be induced by TKI treatment [116,117]. Therefore, treatment leads to changes in tumor vascularity and can lead to a reduction in tumor density, as reflected by the value of the HUs measured on CT (Figure 6). Using RECIST1.1 and Choi, comparable results were obtained for predicting PFS for patients treated with second line sunitinib assessed during an early follow-up of about 2–3 months [96,105]. Nonetheless, the Choi criteria gradually overestimated the number of patients with a partial response to sunitinib and regorafenib during longer follow-up periods (up to a year), leading to poorer PFS [105,106]. It was speculated that a drop in tumor density could also be caused by tumor necrosis, which is often a sign of progressive disease. So, instead of measuring a reduction in tumor vascularization, one may be measuring progressive disease over longer follow-up periods [105].

#### 3.4.2. [^18^F]FDG-PET Imaging

In [^18^F]FDG-PET imaging, the European Organization for Research and Treatment in Cancer (EORTC) PET criteria are most commonly used, in which a metabolic response is determined by a reduction in SUVmax of 25% or more [118]. Metabolic response was significantly associated with prolonged PFS and could be detected as early as seven days, after the induction of TKI treatment (imatinib and sunitinib) [100,102]. On the contrary, the prospective study of Chacón et al. did not find a significant association between PFS and metabolic response determined by the EORTC PET criteria.

Additionally, two retrospective studies by Farag et al. evaluated the impact of [^18^F]FDG-PET/CT on clinical decision making in the treatment of localized and advanced GIST patients. Changes in surgical management, systemic treatment and treatment objective were all included in the evaluation [111,112]. In 27.1% of GIST patients treated with neoadjuvant intent, management was changed because of [^18^F]FDG-PET/CT findings at an interval of eight weeks. The lack of metabolic response was correlated with therapeutic changes in management, especially in non-KIT exon 11 mutations [111]. In the advanced disease setting, specifically late [^18^F]FDG-PET response findings (median of 293 days) proved to have an impact on therapeutic decision [112]. 

#### 3.4.3. CE-CT vs. [^18^F]FDG-PET Imaging

When comparing the aforementioned response evaluation criteria on CE-CT imaging with the EORTC PET criteria on [^18^F]FDG-PET imaging, articles reported high agreement and RECIST responders also showed significant reductions in SUVmax [91,98,100,108]. Choi et al. showed greater sensitivity and specificity (97.0% and 100%) when compared to the EORTC PET criteria [95]. Metabolic response could, however, be observed within a week and preceded changes in tumor size and volume in localized and advanced GIST patients treated with imatinib (Figure 7) [92,97,100,107]. By using the RECIST criteria, the early effect of TKI treatment may be underestimated. For example, Choi et al. showed that 70% of the stable disease RECIST patients had an SUVmax reduction between 61 and 100% at the two-month follow-up [94]. 

## 4. Discussion

The aim of this review was to provide an overview of the value of CE-CT and [^18^F]FDG PET/CT imaging to predict and monitor TKI treatment response in GIST patients.

There is limited literature available on the use of baseline [^18^F]FDG-PET findings to predict tumor response. Although there are only a few studies available, generally imaging features, such as MTV and TLG, were correlated with more aggressive tumor behavior. On the contrary, there is more data available on the potential of CE-CT imaging features to predict treatment response. Results indicate that larger tumor sizes (>5 cm), ill-defined or lobulated tumor outline, mixed or exophytic growth patterns, the presence of (enlarged) and feeding vessels are associated with patient outcome. The presence of heterogeneous enhancement patterns was a recurring observation in high-risk GISTs. The hypodensities observed on CE-CT imaging were devoted to the biological phenomena of neovascularization and necrosis. It should be noted that the correlation between hypodensities on radiological imaging and actual pathological necrosis and neovascularization in GIST tumors is still disputable. 

Many articles discussed the use of radiomic and deep learning models for response prediction on baseline CE-CT imaging. High performance scores were stated for models predicting RFS and risk stratifications, while mutational status remained difficult to predict with variable AUC values. Radiomics offers the possibility to identify clinically relevant imaging features that would normally be imperceptible to the naked eye. For example, it has proven to be difficult to obtain a sufficient amount of tissue samples from biopsy material, which makes it difficult to determine the mutational status or a reliable mitotic count. Additionally, if the mitotic count is determined on postoperative surgical specimens, the results can be inaccurate due to the occasional administration of neoadjuvant TKI treatment. It would, therefore, be very helpful if imaging could provide additional information, other than tumor size. Nonetheless, the biological explanation behind the efficacy of radiomic features was often missing in the included articles. Before advanced and objective learning techniques can be introduced in clinical practice, they should be clinically relevant and biologically meaningful. It is recommended to further explore the prediction of actual radiological response using semantic or quantitative imaging features selected based upon tumor biology. 

The three evaluation methods currently used to monitor response in GIST patients, are the RECIST, Choi and EORTC PET criteria. The main disadvantage of the RECIST criteria is the one-dimensional nature of its measurements, presuming a spherical tumor shape throughout the entire course of TKI treatment. To overcome this limitation, an additional set of criteria was developed by Choi et al. involving CT densities. The Choi criteria are occasionally applied in clinical practice. However, its efficacy and prognostic value in determining response in GIST patients remains unclear. Supposedly, the antiangiogenic effect of TKI treatment would lead to a consequent reduction in HU values. As previously stated, necrosis and heterogeneous enhancement patterns at baseline were considered predictive for more aggressively behaving tumors. Using reductions in CT densities as a criterion for response monitoring may, therefore, be misleading, since it can reflect a decrease in angiogenesis induced by TKI treatment, as well as necrosis induced by aggressive tumor behavior. This hypothesis was supported by literature, since response evaluation using Choi criteria led to an overestimation in the number of partial responders at longer follow-up periods.

[^18^F]FDG-PET proved to be useful in the early monitoring of GISTs, since significant reductions in SUVmax could be observed within a week of TKI treatment and metabolic changes preceded morphological changes in size. However, this imaging technique is often not considered for early response monitoring in clinical practice because of higher costs. Since some of the targeted treatments are more expensive than PET-CT scans, further research should, therefore, be focused on the cost-effectiveness of [^18^F]FDG-PET imaging in the treatment of GISTs. 

Particularly, the combined use of different imaging modalities, also known as multimodality imaging, might provide more detailed information that can assist in making early image-guided treatment decisions. The use of such a multimodality imaging approach might be useful to gather as much information as possible on the biological behavior of GIST. However, currently, no literature is available on the specific use of combining these different imaging modalities for response prediction or monitoring.

## 5. Conclusions

In conclusion, imaging features obtained from CE-CT and [^18^F]FDG PET/CT imaging can aid in the development of a more personalized treatment of GIST patients by enabling early prediction and monitoring of TKI therapy response. Heterogeneous enhancement patterns on baseline CE-CT imaging were predictive for high-risk GISTs, reflecting neovascularization and necrosis.

For the purpose of response monitoring, current RECIST and Choi criteria are still lacking sensitivity and are prone to errors when predicting or monitoring treatment response. [^18^F]FDG-PET is a promising imaging technique that visualizes functional metabolic changes in GISTs, which precedes measurable changes in tumor size. Although promising, the true added value of [^18^F]FDG-PET remains elusive, and research on cost-effectiveness is warranted.

Radiomics is an emerging topic in medicine and shows potential for the prediction of RFS and risk stratifications in GISTs. However, future research should mainly focus on clinical utility, explainability and correlation with actual tumor biology.

## Figures and Tables

**Figure 1 diagnostics-12-02722-f001:**
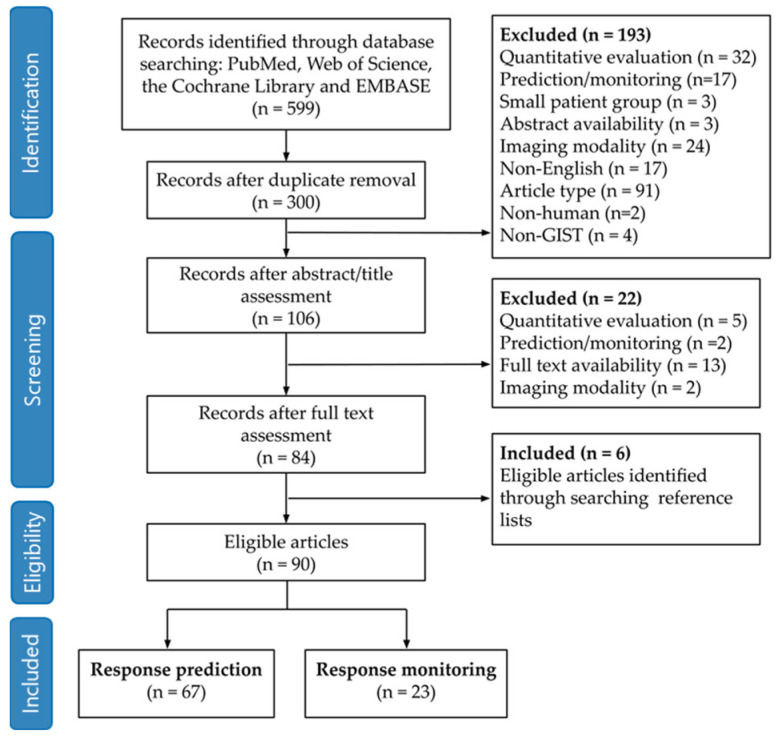
Preferred Reporting Items for Systematic Reviews and Meta-analysis (PRISMA) flowchart showing all the exclusion criteria. A total of 90 articles were included for this systematic review. Sixty-seven articles covered the topic of response prediction, and 23 articles covered the topic of response monitoring.

**Figure 2 diagnostics-12-02722-f002:**
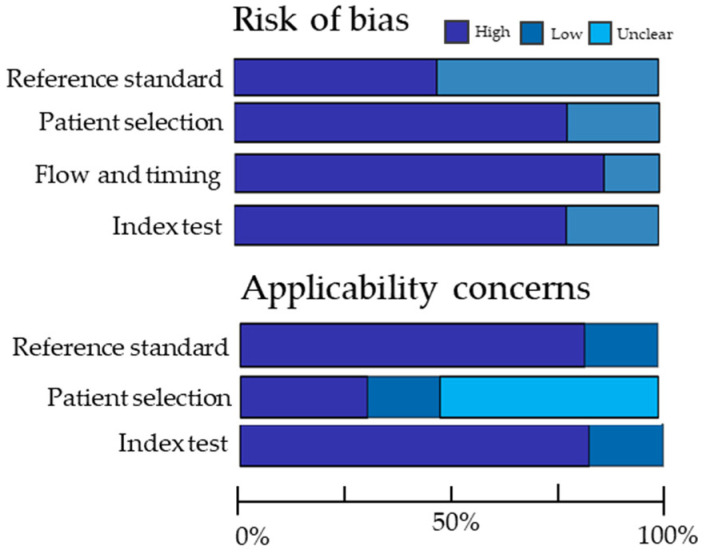
Summary of Methodological Quality Scored According to Quality Assessment of Diagnostic Accuracy Studies Tool, Version 2 (QUADAS-2) for 23 articles discussing the topic of response monitoring.

**Figure 3 diagnostics-12-02722-f003:**
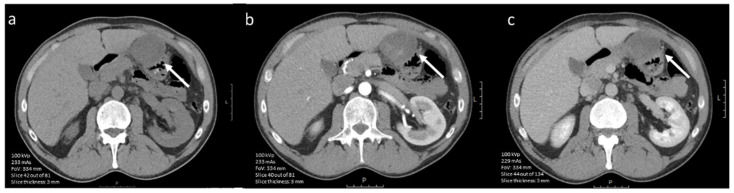
(**a**–**c**). Axial contrast-enhanced (iodinated contrast media) CT image of a 45-year-old male diagnosed with a (histopathologically confirmed) low mitotic gastric GIST. The lesion (arrow) shows a round tumor shape and homogeneous enhancement in (**a**) nonenhanced phase, (**b**) arterial phase and (**c**) portal venous phase (scale bars 5 cm).

**Figure 4 diagnostics-12-02722-f004:**
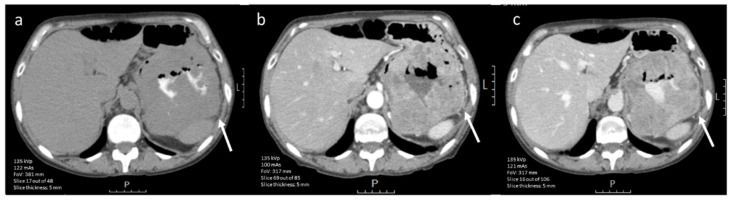
(**a**–**c**). Axial contrast-enhanced (iodinated contrast media) CT images of a 60-year-old male diagnosed with a (histopathologically confirmed) high-mitotic gastric GIST. The lesion (arrow) shows a lobulated tumor shape, heterogeneous enhancement in (**a**) nonenhanced phase, (**b**) arterial phase and (**c**) portal venous phase (scale bars 5 cm).

**Figure 5 diagnostics-12-02722-f005:**
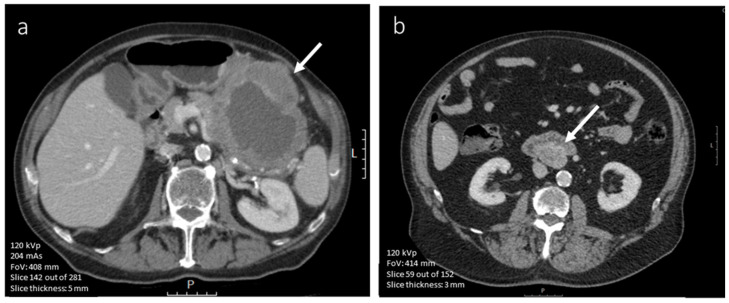
(**a**,**b**). Axial portal venous phase (iodinated contrast media) CT image of an 83-year-old male diagnosed with a high-risk (Miettinen AFIP classification) gastric GIST (scale bar 5 cm). The large lesion is lobulated and has central necrosis (arrow). (**b**) Axial portal venous phase (iodinated contrast media) CT slice of a 72-year-old male diagnosed with a low-risk GIST affecting the small intestine (scale bar 5 cm). It shows a well-defined and rounded lesion with a homogeneous enhancement pattern (arrow).

**Figure 6 diagnostics-12-02722-f006:**
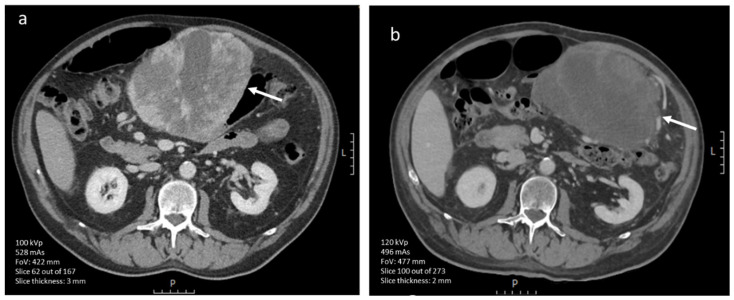
(**a**,**b**). Axial portal venous phase CT images (iodinated contrast media) of a 67-year-old male diagnosed with a primary GIST (arrows) of the stomach. (**a**) Pre-treatment imaging shows a large gastric mass with heterogeneous enhancement. (**b**) After 1.5 months of avapritinib treatment, the lesion has become hypodense (scale bars 5 cm).

**Figure 7 diagnostics-12-02722-f007:**
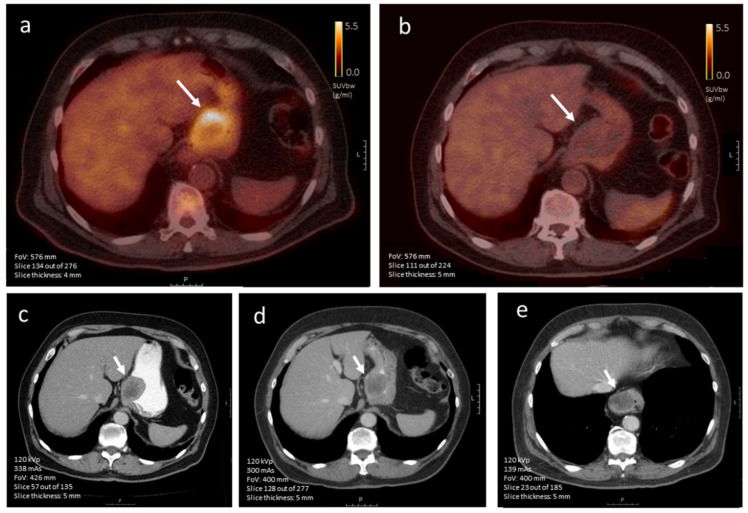
(**a**–**e**). Axial [^18^F]FDG-PET images of a 71-year-old male diagnosed with a primary gastric GIST (**a**) before treatment induction and (**b**) after about three months of TKI treatment, where the standardized uptake value (SUV) is normalized (scale bar 5 cm). Corresponding contrast-enhanced CT imaging (iodinated contrast media) visualizing the same lesion (arrow) (**c**) at diagnosis and after (**d**) 2.5 months and (**e**) 6.5 months of imatinib treatment, showing minimal to no change in tumor size (scale bar 5 cm). In the last image, the intrathoracic tumor location is caused by a sliding hernia.

## Data Availability

The datasets generated during and/or analyzed during the current study are available from the corresponding author upon reasonable request.

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
