# Peer review of "Early Prediction and Monitoring of Treatment Response in Gastrointestinal Stromal Tumors by Means of Imaging: A Systematic Review"

_diagnostics, 2022, doi:10.3390/diagnostics12112722_

Round 1
Reviewer 1 Report
Very interesting and well written paper about the role of CE-CT and PET/CT in predicting the response to TKI in GIST.
I endorse its publication in current form.
Author Response
I would like to thank you for your positive reviews on the manuscript on response prediction and monitoring of treatment response in gastrointestinal stromal tumours.
Reviewer 2 Report
This is an excellent article, but I have comments mentioned below:
1- what is the source of the figures? Is it from the authors' cases or other sources, and is there permission for publication?
2- all figures from the stomach; is this indicative of particular interest?
3- please add scale bar, annotations, insets, type of dye or stain, and magnifications to all figures
Author Response
Dear reviewer,
Firstly, I would like to thank you for reviewing the manuscript on response prediction and monitoring of treatment response in gastrointestinal stromal tumours. This cover letter encloses a point-by-point response to the comments and suggestions.
Point 1: What is the source of the figures? Is it from the author’s cases or other sources, and is there permission for publication?
Response: All figures were retrieved from the Leiden University Medical Center GIST imaging database. Images were not retrieved from other sources and therefore no permission from copyright holders was needed. This study was executed under the protocol “Multimodality imaging for improving personalized care in sarcoma patients” (protocol code: B19.050). This protocol was approved by the medical ethics committee Leiden Den Haag Delft (METC LDD). Declaration of no-objection was issued on the 14th of January. Patient confidentiality was guaranteed through anonymized and untraceable data. To clarify, we will add this explanation at the end of the manuscript, under the section “Institutional Review Board Statement”.
Point 2: All figures from the stomach; is this indicative of particular interest?
Response: This is not necessarily indicative for a particular interest, but rather coincidental. GISTs affect the entire gastrointestinal tract, but most commonly the stomach and the small intestine. Rectal GISTs are routinely monitored through Magnetic Resonance Imaging, which is an imaging modality that falls outside of the scope of this review. In addition, the authors wanted high quality imaging examples to illustrate the described phenomena. The most suitable examples were mainly found in gastric (Fig. 3, 4, 5a, 6 and 7) and duodenal (Fig. 5b) GISTs.
Point 3: Please add scale bar, annotations, insets, type of dye or stain, and magnifications to all figures.
Response: The scale bar, annotations, insets, type of dye or stain, and magnifications will be added to all figures. Annotations such as the hospital name and the date of examination will not be included for privacy reasons.